# Identification of factors associated with morbidity and postoperative length of stay in surgically managed chronic subdural haematoma using electronic health records: a retrospective cohort study

Daniel J Stubbs [1,2] Benjamin M Davies [3] Tom Bashford [1,2,4]
Alexis J Joannides,[5,6] Peter J Hutchinson,[6] David K Menon [1,4,7] Ari Ercole [1,7]
Rowan M Burnstein[7]

For numbered affiliations see end of article.

**Correspondence to**
Dr Daniel J Stubbs;
djs225@cam.ac.uk

## ABSTRACT

**Introduction** Chronic subdural haematoma (cSDH) tends to occur in older patients, often with significant comorbidity. The incidence and effect of medical complications as well as the impact of intraoperative management strategies are now attracting increasing interest.

**Objectives** We used electronic health record data to study the profile of in-hospital morbidity and examine associations between various intraoperative events and postoperative stay.

**Design, setting and participants** Single-centre, retrospective cohort of 530 cases of cSDH (2014–2019) surgically evacuated under general anaesthesia at a neurosciences centre in Cambridge, UK.

**Methods and outcome definition** Complications were defined using a modified Electronic Postoperative Morbidity Score. Association between complications and intraoperative care (time with mean arterial pressure <80 mm Hg, time outside of end-tidal carbon dioxide ($ETCO_2$) range of 3–5 kPa, maintenance anaesthetic, operative time and opioid dose) on postoperative stay was assessed using Cox regression.

**Results** 53 (10%) patients suffered myocardial injury, while 24 (4.5%) suffered acute renal injury. On postoperative day 3 (D3), 280 (58% of remaining) inpatients suffered at least 1 complication. D7 rate was comparable (57%). Operative time was the only intraoperative event associated with postoperative stay (HR for discharge: 0.97 (95% CI: 0.95 to 0.99)). On multivariable analysis, postoperative complications (0.61 (0.55 to 0.68)), anticoagulation (0.45 (0.37 to 0.54)) and cognitive impairment (0.71 (0.58 to 0.87)) were associated with time to discharge.

**Conclusions** There is a high postoperative morbidity burden in this cohort, which was associated with postoperative stay. We found no evidence of an association between intraoperative events and postoperative stay.

### Strengths and limitations of this study

► First use of electronic health data to chart the time course of in-hospital complications in a neurosurgical cohort.
► Use of multiple imputation to maximise power with clear comparison to complete case analysis for comparison.
► Methodological approaches are scalable to other surgical and in-patient cohorts.
► Main results limited by being a single-centre, retrospective study, and thus lack external validity.

## INTRODUCTION

Chronic subdural haematoma (cSDH) is a common neurosurgical pathology of encapsulated blood products beneath the dural membrane, driven by trauma and inflammation.[1] Although cSDH can be managed conservatively, surgical evacuation is favoured in the presence of neurological deficit. cSDH is predominantly a 'disease of the elderly'. It is also associated with frailty and comorbidity.[2–5] Increased mortality at 1 year among individuals with a cSDH has led some to consider it a 'sentinel health event'.[6] In these respects, there is a clear analogy with patients suffering a fracture of the neck of femur (NOF), a group whose outcomes have recently been improved by concerted efforts into the measurement, integration and improvement of holistic perioperative care.[7 8] However, with cSDH, although a similarly effective surgical intervention exists, there has, as yet, been no demonstration of a systematic improvement in patient outcomes through analogous process changes.

To date, the impact of non-neurological morbidity in patients with cSDH has received little attention.[9] The reasons for this are likely multifactorial, but crucially there are significant challenges in adequately capturing patient morbidity using standard methodological approaches, which typically rely on cross-sectional assessment at set time points.[10] However, there is some evidence to suggest that non-neurological morbidity may be a significant problem. In a nationwide audit of UK cSDH management, 110 (14%) of 787 surgically treated patients suffered a complication, of which pneumonia was the most common.[11] The prevalence of such complications appears to increase with age.[12 13] The experience from parallel surgical fields would suggest this morbidity has implications for both patient outcomes, and the delivery of cost-effective healthcare.[14] This highlights a need for a more detailed investigation into the relationship between perioperative morbidity and outcome in cSDH. An understanding of the impact of all stages of the perioperative pathway on patient outcome is vital if potential clinical and service benefits from tailored process change, analogous to those in the NOF population, are to be realised.[8]

Surgery for cSDH is commonly performed under general anaesthesia (GA). Heterogeneity in anaesthetic practice has the potential to introduce variation in postoperative outcome, yet the impact of intraoperative events on outcome has received little attention in cSDH. Physiological changes, such as hypotension, can be a direct pharmacodynamic consequence of GA. Importantly, even brief periods below a mean arterial pressure (MAP) of 80 mm Hg are associated with end-organ damage,[15] and it is reasonable to assume that the cSDH population may be particularly vulnerable. Furthermore, arterial carbon dioxide tension ($PaCO_2$) is no longer under homeostatic control during mechanical ventilation, which may have important implications for cerebral blood flow (CBF) and intracranial pressure (ICP).[16] In cSDH, changes in CBF correlate with clinical symptoms.[17] GA choice may also have postoperative effects. Opioids have numerous side effects that occur with increasing frequency in the elderly.[18] The varied pharmacodynamic effects of anaesthetic agents have the potential to differentially impact on both ICP and CBF,[19] with intravenous anaesthesia associated with lower rates of certain complications such as nausea and vomiting.[20] However, a recent systematic review (of low quality evidence) failed to demonstrate a difference in delirium rate or length of stay (LOS) between patients receiving a volatile or intravenous anaesthetic.[21]

The advent of integrated electronic health records (EHR) allows the capture of complex events, such as intraoperative physiology, and to comprehensive measurement of in-patient morbidity across the entirety of an admission. New scoring systems are required to generate computable phenotypes from such complex, routinely collected data.[22] One such score, an electronic variant of the widely used Postoperative Morbidity Score (POMS),[10] has equivalent discriminative performance in identifying discharge complexity or prolonged LOS in a heterogeneous population of elderly surgical patients.[22] LOS is often critiqued as an outcome measure in improvement efforts due to the impact of non-clinical events.[23] The complex relationship between clinical and non-clinical events in determining LOS is likely to vary between patients with differing disease processes. Regardless, its utility from the perspective of hospital bed occupancy is inarguable, with mounting health service pressures bed utilisation and patient throughput with minimal accrued morbidity is of crucial importance.

The aim of this study was to use routinely collected EHR data to examine, for the first time, the impact of intraoperative events and medical complications on postoperative length of neurosurgical centre stay in patients undergoing surgery for cSDH.

## METHODS

### Cohort selection

This was a single-centre, retrospective evaluation of surgically treated cSDH at Cambridge University Hospitals NHS Foundation Trust (CUH) between 26 October 2014 and 5 January 2019. CUH offers neurosurgical and trauma services to a population of approximately 5.8 million in the East of England (online supplementary figure S1).[24] Eight hundred and nine cases of cSDH were identified from the centre's neurosurgical referrals database and operation log, coded contemporaneously by neurosurgeons. After excluding missing records, duplicates and incorrect procedures, 530 patients were included in this retrospective cohort (online supplementary figure S2). The majority of patients (491, 92.5%) were transferred from other hospitals (n=14). Anonymised data for each case were extracted from the CUH EHR (Epic Systems Corporation, Verona, Wisconsin, USA) by the centre's clinical informatics team.

### Patient and public involvement statement

Due to the retrospective and evaluative nature of the project, no formal patient and public involvement work was undertaken.

### Local surgical practice

Local practice is relatively standardised and comparable to the previously published literature.[25] Procedures are predominantly performed under GA via two burr holes with subsequent irrigation and usage of a subdural drain on free drainage. The drain is removed after 48 hours with patients referred back to their local hospital at this point if required. Standard postoperative care is performed on a designated neurosurgical ward. Physiological observations are measured at a frequency dependent on the patients calculated 'Early Warning Score' in line with UK national guidance[26] and at least two times within a 24-hour period for all inpatients. Mini-craniotomy is reserved for recurrence or complex collections. Due to the use of standard Classification of Interventions and procedure

**Table 1** Definitions of the adapted Electronic Postoperative Morbidity Score (ePOMS) used for daily assessment of complications in a retrospective cohort of operated chronic subdural haematoma

| Domain | Diagnostic criteria | Notes |
|---|---|---|
| Respiratory | Need for supplementary oxygen | |
| Cardiovascular | HR >100<br>SBP <100<br>Positive troponin test | |
| Neurological | Need for 1:1 nursing observation<br>Motor/Verbal Score worse than referral<br>Focal neurology | Surrogate for hyperactive delirium<br>Documented mismatch between left/right sided arm/leg strength at any stage |
| Renal | Rise in creatinine to ≥1.5× baseline | Last recorded creatinine prior to surgery |
| GI | Administered antiemetic | Defined by the following WHO ATC codes: a04*, a03fa01, r06ae03, n05ad08 |
| Pain | Need for intravenous opioids or local anaesthetic infusion on a given day | Drugs identified by following WHO ATC codes: n02aa01, n02ab03, n01bb01 |
| Recurrence† | Reoperation | ePOMS originally identifies severe wound infection by need for further surgery. In this context reoperation for the same procedure is being used to identify reaccumulation of cSDH |
| Infection | Temperature ≥38°C<br>Receiving antibiotics on a given day | Antibiotics defined by following WHO ATC codes: j01‡ |
| Haematological | Transfused with blood product | Including red cells, platelets, FFP, cryoprecipitate |

If multiple potential criteria are listed then an individual is positive in that domain if any of these are met.
*Indicates additional criterion included in this variant of ePOMS from those previously published.
†In the original ePOMS this would correspond to the wound category.
‡Indicates that all drugs below this level of ATC code were included. A dictionary of relevant ATC codes is available in the online supplementary material.
FFP, fresh frozen plasma; GCS, Glasgow Coma Scale; GI, gastrointestinal; HR, heart rate; SBP, systolic blood pressure; WHO ATC, World Health Organization anatomical therapeutic chemical classification.

(OPCS) codes for all forms of subdural drainage, specific surgical approach was unable to be resolved.

### Covariate definitions
#### Perioperative medical complications
A variant of the electronic POMS (ePOMS)[22] was used to determine the presence of morbidity across nine distinct domains, following key organ systems (respiratory, cardiovascular, neurological, renal, haematological, gastrointestinal, infectious, reoperation/wound and pain) (table 1). Recognising that ePOMS had not previously been employed in a neurosurgical population, the score was modified to capture a greater number of dimensions of neurological morbidity. These included a deterioration in motor or verbal components of the Glasgow Coma Scale (GCS) (from admission), and the presence of asymmetric motor power. Therefore, the total score for an individual on any given day ranged between 0 and 11. Electronic observation, drug and laboratory result charts were extracted from the EHR and the daily ePOMS generated for each patient. Physiological criteria (eg, heart rate) were fulfilled if two or more instances occurred on a given day compared with a single event for investigation and drug-based criteria.

#### Intraoperative covariates
Automatically recorded physiological measurements (blood pressure and heart rate) and inspired/expired gas measurements were extracted from the EHR at minute resolution throughout GA. Blood pressure was measured both non-invasively (typically at a frequency of more than every 5 min[27]) and, for a subset of patients, invasively from an arterial catheter. All MAP values were combined and linearly interpolated. Total time (in minutes) spent below a threshold of 80 mm Hg[15] was calculated. In an effort to reduce artefact, values of 0 or >200 mm Hg were excluded. Maintenance anaesthetic was deemed inhalational if there were non-zero values of an inhalational anaesthetic agent. Text labels of manually recorded drug administrations were converted using a lookup table matching drugs to their corresponding WHO anatomical therapeutic classification (WHO-ATC) code.[28] All administrations of a drug with a code of N02A (opioid class) were extracted along with their dose. All opioid administrations were expressed as fentanyl equivalents using published conversions[29] as this is the most commonly used opioid in local neuroanaesthetic practice. Brain injury guidelines suggest that ventilation is adjusted to keep the level of carbon dioxide measured in arterial blood ($PaCO_2$) less than 6 and above 3.5 kPa.[30] As end tidal values underestimate the true arterial gas tension by between 0.5 and 1 kPa, time above an end-tidal carbon dioxide ($ETCO_2$) of 5 kPa and time below an $ETCO_2$ of 3 kPa was calculated to give a total time outside of optimal $CO_2$ range.

## Other covariates

To control for residual confounding, key admission characteristics were also extracted from the EHR and referral database. These included: age, sex, American Society of Anesthesiologists (ASA) Score, Modified Rankin Scale (mRS), admission location (direct or from a referring hospital), baseline neurological status (GCS 15 yes/no and Motor Score 6 yes/no), admission creatinine and key comorbidities. Preoperative deterioration (defined as a worsening ePOMS between admission and time of surgery) and length of wait for surgery were also controlled for. Cognitive state on admission was defined using an Admission Screening Questionnaire that ascertains the presence of temporary or permanent cognitive impairment. This was collapsed into a binary definition of 'any' cognitive impairment versus none.

As the majority of patients referred for surgery had no previous admission on record, the baseline list of comorbidities was frequently incomplete. Comorbidities were inferred from admission medications using the 'Rx-Risk' Score, which uses WHO-ATC codes to determine the presence of specific comorbidities.[31] This approach was chosen over discharge clinical coding to avoid potential reverse causation (eg, in-hospital myocardial infarction (MI) being wrongly interpreted as a pre-existing history of MI). Preadmission anticoagulant/antiplatelet use, airway disease, heart failure or cardiovascular disease was determined using medication records reconciled in the first 24 hours of an individual's admission. Full codes used to determine comorbidity are shown in online supplementary table S1.

## Statistical techniques

All analyses were performed in R (V.3.5.3).[32] The effect of covariates on postoperative LOS was performed using Cox regression. ePOMS values were included as a time-dependent covariate calculated per day, and all other variables were assumed to exert effect over the entire postoperative period. Univariable analysis was performed for all variables, those with a $p < 0.1$ on initial regression were carried forward for multivariable modelling. At this stage, statistical significance was taken as $p < 0.05$. All analyses were conducted in both complete case and multiply imputed data sets.

## Handling of missing data

Daily entries in the EHR for calculating ePOMS (such as blood results, observations and administered drugs) were assumed to be accurate (eg, if no troponin test was ordered, a value was not imputed). Anaesthetic chart data (observations, measurements of gas tension) were also taken to be accurate due to automated data capture. Baseline patient variables (such as ASA Score) were imputed using multiple imputation with chained equations.[33] This was performed using the R packages 'mice'[34] and 'naniar'.[35] A full list of R packages used is available in the online supplementary material. A multilevel imputation model, clustered by individual, was used to allow imputation of baseline data alongside longitudinal ePOMSs.[36] Visualisation of missing data patterns are shown in online supplementary figure S3. Four variables had missing data; baseline creatinine (missing in 46%–8.7%), cognitive status (36%–6.8%), ASA Score (87%–16.1%) and mRS (192%–36.2%). All four were missing in only two cases. Forty imputed data sets were generated. Comparison of imputed and observed values can be seen graphically in online supplementary figure S4. All results presented in the results reflect pooled analyses across all imputed data sets, corresponding complete case analyses are in the online supplementary tables S2,S3.

# RESULTS

## Cohort characteristics

These are summarised in table 2, dichotomised by referral source. Direct admissions were younger (median age 74 vs 77, $p = 0.010$), and had fewer patients with a favourable Motor Score on admission (79.5% vs 93.5%, $p = 0.001$). Time to surgery after admission to the tertiary centre was not significantly different but this did not account for any additional wait in referring hospitals (median=10.6 (4.7–28.4) hours). Reoperation for recurrence was associated with significantly longer time to discharge (HR 0.38 $p < 0.0001$).

## Operative exposures

Two hundred and thirty-five (44.2%) individuals received a volatile anaesthetic. Two hundred and forty (45.2%) had a recorded entry of an intravenous anaesthetic. The majority of remaining cases likely received intravenous anaesthetic (due to the frequent presence of other anaesthetic drugs such as neuromuscular blockers on their drug chart) but could include a minority of cases performed under local anaesthetic.

Median duration of surgery was 96 min (IQR: 78.0–103.5) with a maximum length of 318 min. Median time with a MAP <80 mm Hg was 45 min (IQR: 25–69). Median time outside of the optimal $CO_2$ range was 8 min (IQR: 4–15). Median opioid dose in fentanyl equivalents was 100 µg (IQR: 50–150).

## Inpatient morbidity

Thirteen individuals (2.4%) died during admission. Forty-nine (9.2%) patients were re-operated for their cSDH, twenty-eight during their index admission. Fifty-three (10%) individuals exhibited some degree of postoperative troponin rise, with twenty-four (4.5%) developing acute kidney injury. Three hundred and forty-six (65.2%) patients had a degree of preoperative deterioration between admission and surgery as determined by an increase in their ePOMS between these time points.

The prevalence of each complication subtype shown by postoperative day is demonstrated in figures 1 and 2. There was an obvious restoration in focal neurology over the first 2 postoperative days (figure 1), although no such trend was apparent in those needing 1:1 observation.

**Table 2** Baseline characteristics of retrospective cohort of operated chronic subdural haematoma (n=530)

| | All (n=530) | Referred (n=491) | Local (n=39) | P value |
|---|---|---|---|---|
| **Variable** | **Median (IQR)** | | | |
| Age, years | 77 (69–84) | 77 (70–84) | 74 (62–79) | 0.010* |
| Creatinine, µmol/L | 73 (61–89) | 73 (61–89) | 76 (64–100) | 0.350 |
| Time to surgery (within centre), hours | 20.1 (9.5–40.3) | 20.1 (9.8–42.7) | 19.7 (5.2–35.9) | 0.363 |
| | **n (%)** | | | |
| Male | 376 (70.8) | 349 (70.9) | 27 (69.2) | 0.855 |
| ASA ≥3 | 271 (61.0)% | 250 (61.2) | 20 (57.1) | 0.718 |
| Cognitively impaired | 270 (54.5)% | 254 (54.9) | 16 (48.5) | 0.485 |
| Admission GCS 15 | 342 (64.4) | 317 (64.5) | 25 (64.1) | 1 |
| Admission Motor Score 6 | 490 (92.3) | 459 (93.5) | 31 (79.5) | 0.001* |
| mRS ≥2 | 105 (31.0)% | 104 (21.2) | 1 (20.0) | 1 |
| Anticoagulants/antiplatelets | 232 (43.9) | 211 (43.0) | 21 (53.8) | 0.241 |
| CVS disease | 238 (45.0) | 225 (45.8) | 13 (33.3) | 0.136 |
| Heart failure | 101 (19.0) | 90 (18.3) | 11 (28.2) | 0.139 |
| Airways disease | 75 (14.1) | 66 (13.4) | 9 (23.1) | 0.099 |

'Referred' indicates cases referred from a regional hospital for surgery. 'Local' indicates a case admitted directly to the tertiary centre. Motor Score refers to score on the motor (movement) component of the Glasgow Coma Scale.
ASA of 3 or more indicates presence of significant systemic comorbidity, Modified Rankin Scale of 2 or more indicates a level of disability that impedes normal activity. % Indicates that value is calculated only on those with recorded values. P value calculated for Mann-Whitney U (continuous data) or Fisher's exact test (categorical).
*Indicates significant at 5% level.
ASA, American Society of Anesthesiologists Score; CVS, cardiovascular; GCS, Glasgow Coma Scale; mRS, Modified Rankin Score.

Plots of incident complications in neurological, cardiovascular, respiratory, gastrointestinal, pain and infectious complications by day are shown in online supplementary figures S5 and S6. Cases of new, worsening neurology after surgery were rare (online supplementary figure S5), a pattern which differed from that of non-neurological

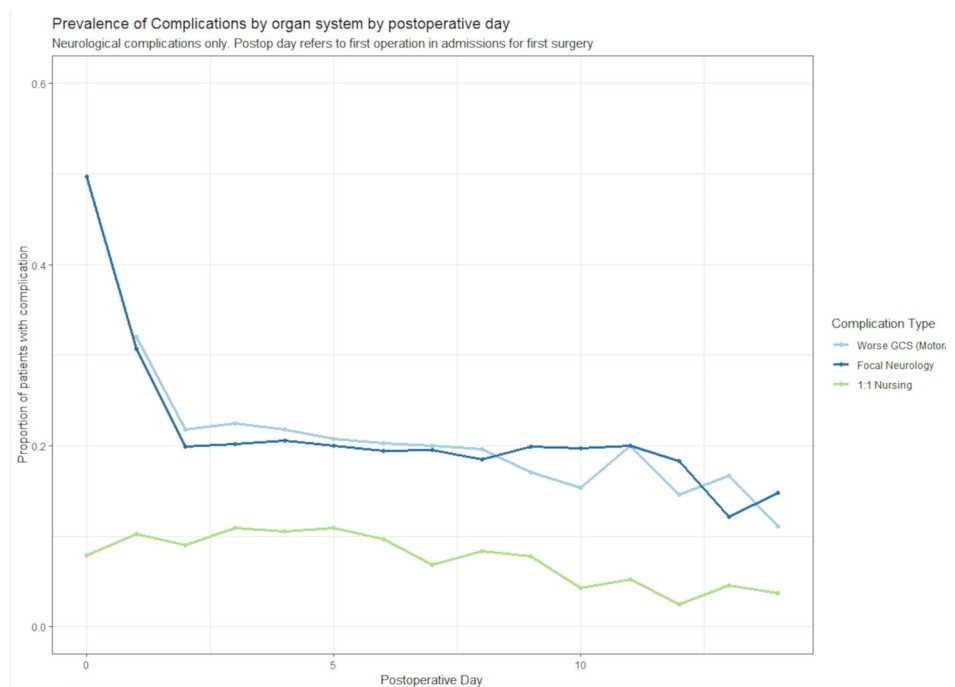

**Figure 1** Proportion of inpatients with specific neurological complications, extracted from an electronic health record, after surgery for chronic subdural haematoma, by postoperative day (n=530). GCS, Glasgow Coma Scale.

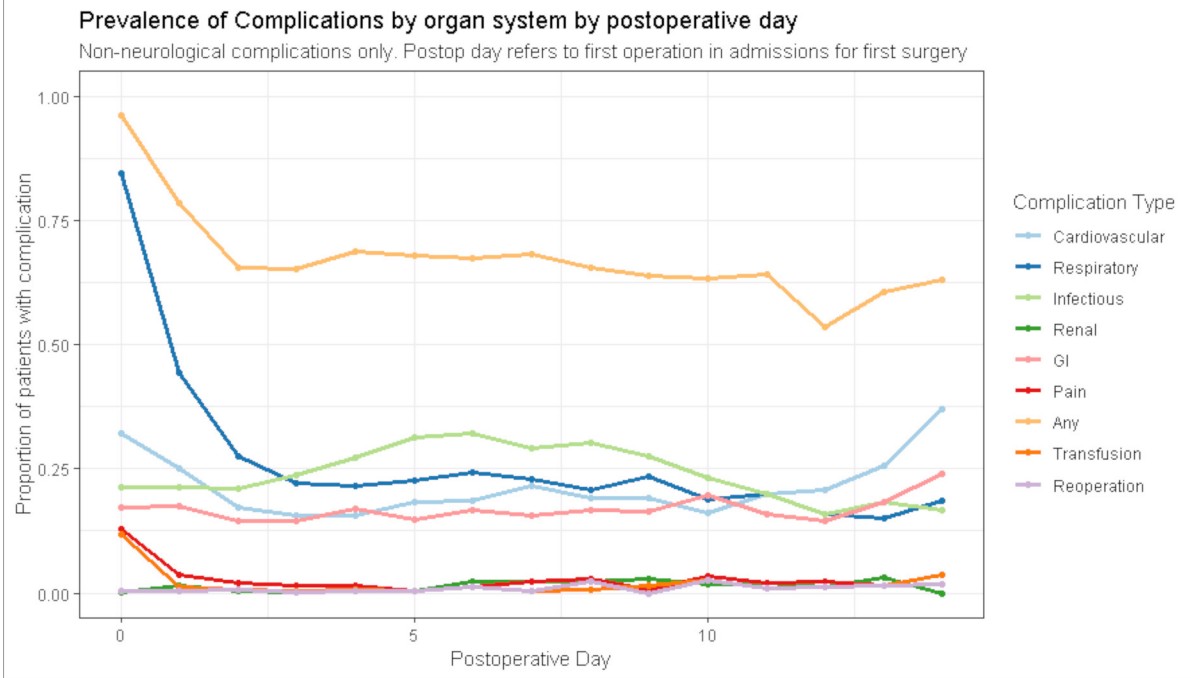

**Figure 2** Proportion of inpatients with organ specific complications (as defined by the Electronic Postoperative Morbidity Score), after surgery for chronic subdural haematoma, by postoperative day (n=530). GI, gastrointestinal.

complications (especially cardiovascular and respiratory) where new cases occurred throughout measured inpatient stay.

The apparently high prevalence of respiratory 'complications' on day of surgery and postoperative day 1 is likely spurious, reflecting the routine use of supplemental oxygen to counteract postanaesthetic effects. The prevalence of infective complications appeared highest between days 3 and 10 (figure 2). By day 3, 280 individuals (58% of 484 remaining inpatients) had at least one complication, a comparable rate to day 7—138 of 242 remaining inpatients (57%).

### Length of stay

Overall median LOS in the tertiary centre was 6.7 (IQR: 4.2–10.6) days. For directly admitted patients, this was 8.9 (IQR: 5.2–16.3) days versus 6.5 (IQR: 4.1–10.1) days for those referred from elsewhere. Postoperative LOS was 4.9 (3.0–8.8) days in referred patients, compared with 7.9 (3.4–15.5) days in local patients. Postoperative time to discharge, dichotomised by referral source is shown in figure 3.

### Association of intraoperative events and complications on postoperative LOS.

From examined operative variables, only operative time demonstrated a significant association (HR: 0.97 (0.95–0.99)) on univariable analysis. Postoperative complications were associated with prolonged postoperative stay (HR: 0.57 (0.52–0.63)) when included as a time-dependent covariate. Baseline variables including age, sex, ASA Score, mRS, medical state and comorbidities were associated with longer time to postoperative

discharge (table 3), while a favourable GCS and being from a referred hospital was associated with a more rapid discharge from the tertiary centre.

### Multivariable analysis

Eighteen of these variables were taken forward to multivariable modelling (table 4). This was done to maximise control of confounding. Postoperative complications (HR: 0.61 (0.55–0.68)), cognitive concerns (HR: 0.71 (0.58–0.87)) and anticoagulation (HR: 0.45 (0.37–0.54)) were all associated with longer time to discharge. Admission GCS (HR: 1.03 (0.97–1.08)) and referral source (HR: 1.46 (0.98–2.17)) were not significant in the final model. On complete case analysis, the pattern was similar except for an identified association with more rapid discharge with higher admission complication (ePOMS) scores (HR complete case 1.14 (1.04–1.26) vs HR MI: 1.06 (0.99–1.12)) (online supplementary table S3).

Based on these results, two sensitivity analyses were performed to attempt to understand the apparent lack of association between admission GCS and postoperative stay. First, total GCS was replaced with an indicator for motor score (6 or not) (online supplementary table S4), and second, a potential mediating effect of postoperative complications was examined by removing this covariate (online supplementary table S5). In the first analysis, motor score was not associated with postoperative time to discharge. In the second analysis, no significant association with GCS was revealed in the subsequent model, although significant associations with admission ASA Score, renal function and age did become apparent (online supplementary table S5).

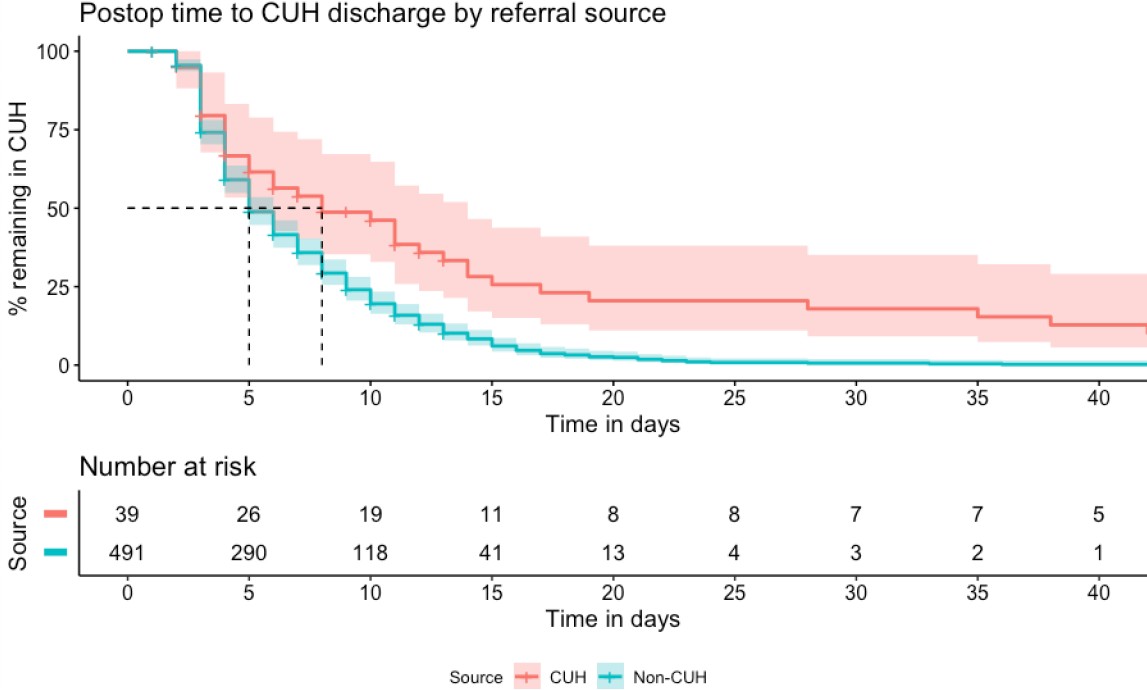

**Figure 3** Kaplan-Meier curves for postoperative time to discharge, dichotomised by referral source, in a retrospective cohort of operated chronic subdural haematoma (n=530). CUH, patients admitted directly to the neurosurgical centre at Cambridge University Hospitals NHS Foundation Trust; Non-CUH, patients referred from other hospitals in the region. Shaded areas=95% CI, dashed lines indicate median postoperative length of stay by group.

## DISCUSSION

To our knowledge, this is the first study in cSDH to use routinely collected EHR data to define postoperative morbidity and clinical interventions, such as anaesthetic events, while exploring their impact on LOS.

### Principal findings

We highlight a significant degree of postoperative morbidity in surgically treated cSDH. Despite comparable cSDH recurrence rates of 9.2% (9% in a UK wide audit of practice[11]), in-patient morbidity captured by the ePOMS demonstrated a prevalence of medical complications of over 50% in the 2 weeks after surgery. Overall, 13% of patients demonstrated a rise in serum troponin or an episode of renal injury. Postoperative morbidity was significantly associated with longer tertiary centre LOS, raising the question of whether efforts to pre-empt or aggressively manage such morbidity could afford more efficient healthcare delivery and improved patient outcome. The total in-hospital burden of morbidity is likely higher than our estimate, as the single-centre nature of our study did not include complications occurring in a patient's referring hospital. Our study design also prevents us examining the associations with total inpatient stay due to likely significant unmeasured events, including bed and transport availability. This data structure also precludes an assessment of mid-term mortality as records are not linked. Regardless, the presented findings are likely to be highly relevant to practice within tertiary neurosurgical centres.

### Baseline factors and confounding

Cognitive function on admission, admission anticoagulation and postoperative complications were all associated with longer time to discharge. Importantly, admission GCS was not associated with postoperative stay. One interpretation could be that any impact of preoperative neurology on postoperative course may be mediated through the included postoperative complication term (eg, low GCS is associated with LOS through a predisposition to complications, such as pneumonia). This explanation was not supported by a sensitivity analysis (online supplementary table S5). A larger question is to consider which measure of neurological morbidity is best for prognostication in this patient cohort, both in terms of underlying pathophysiology and data coding. For instance, an individual with a hemiparesis arising from a non-dominant hemisphere lesion would likely be recorded as possessing a motor score of 6 and a GCS of 15, thus weakening any observed association. Unpicking these issues and the co-linearity, inherent within these overlapping variables is difficult and should be explored in larger data sets.

We have attempted to control for many important variables identified in previous work in traumatic neurosurgical patients.[37] . The shift in HR for referral source between univariable and multivariable modelling suggests that at least some of its observed association with time to discharge reflects confounding or mediation by included variables. This perhaps reflects differences in patient cohorts (such as admission motor score or age—table 2), or management between local and referred patients.

Table 3  Results from univariable Cox regression of effects of specific variables on postoperative length of stay in a cohort of patients undergoing surgery for cSDH (n=530) pooled across m=40 imputed data sets

| | HR | 95% CI | P value |
|---|---|---|---|
| **Demographics** | | | |
| Age (per year increase) | 0.99 | 0.99 to 0.99 | 0.042* |
| Sex (male vs female) | 1.20 | 0.99 to 1.45 | 0.053* |
| Referred patient | 2.20 | 1.54 to 3.13 | <0.001* |
| ASA 2 (all vs ASA 1) | 0.50 | 0.31 to 0.82 | 0.006* |
| ASA 3 | 0.34 | 0.21 to 0.55 | <0.001* |
| ASA 4 | 0.26 | 0.15 to 0.45 | <0.001* |
| ASA 5 | 0.22 | 0.07 to 0.68 | 0.009* |
| mRS 1 (all vs mRS 0) | 0.86 | 0.67 to 1.11 | 0.249 |
| mRS 2 | 0.83 | 0.58 to 1.18 | 0.299 |
| mRS 3 | 0.73 | 0.47 to 1.11 | 0.142 |
| mRS 4 | 0.68 | 0.46 to 0.98 | 0.040* |
| **Admission status** | | | |
| GCS 15 on admission (vs any other) | 1.11 | 1.06 to 1.16 | <0.001* |
| Admission ePOMS (per 1 domain increase) | 0.91 | 0.87 to 0.97 | 0.002* |
| Cognitive impairment (yes/no) | 0.57 | 0.48 to 0.69 | <0.001* |
| **Comorbidities** | | | |
| Airways disease | 0.75 | 0.58 to 0.96 | 0.020* |
| CVS disease | 0.83 | 0.70 to 0.99 | 0.040* |
| Heart failure | 0.70 | 0.56 to 0.87 | 0.001* |
| Anticoagulated on admission | 0.42 | 0.35 to 0.50 | <0.001* |
| Creatinine (per 20 µmol/L increase) | 0.96 | 0.92 to 0.99 | 0.022* |
| **Day of surgery** | | | |
| Preoperative deterioration (yes/no) | 1.07 | 0.89 to 1.28 | 0.488 |
| Length of wait (per hour) | 0.96 | 0.92 to 1.01 | 0.142 |
| Time MAP <80 mm Hg (per 10 min increase) | 0.99 | 0.97 to 1.02 | 0.720 |
| Time $ETCO_2$ not 3–5 kPa (per 10 min increase) | 0.98 | 0.92 to 1.05 | 0.616 |
| Fentanyl dose (per 25 µg increase) | 1.01 | 0.99 to 1.04 | 0.330 |
| Volatile maintenance | 0.99 | 0.84 to 1.18 | 0.964 |
| Operative time (per 10 min increase) | 0.97 | 0.95 to 0.99 | 0.004* |
| **Postoperative** | | | |
| Complications (per 1 domain increase in ePOMS) | 0.57 | 0.52 to 0.63 | <0.001* |

HRs >1 indicate a more rapid time to postoperative discharge.
*Indicate statistical significance at p<0.1, chosen threshold for inclusion in subsequent multivariable modelling. Postoperative complications included as a time-dependent covariate (by day).
ASA, American Society of Anesthesiologists Score; CVS, cardiovascular; ePOMS, Electronic Postoperative Morbidity Score; $ETCO_2$, end-tidal carbon dioxide measurement; GCS, Glasgow Coma Scale; MAP, mean arterial pressure; mRS, Modified Rankin Scale.

These findings require further evaluation in future studies and distinct cohorts.

### Comparison to other literature

Although not widely examined, studies that have examined medical complications in cSDH have suggested that they may be a significant burden. A 2014 cross-sectional survey of UK neurosurgical centres demonstrated that approximately 110 (14%) of 787 surgically treated cSDH suffered a complication. In this survey, morbidity was defined as, '…any adverse event occurring during inpatient stay…',[38] with a predefined audit target of <10%. In our study, ePOMS defined morbidity peaked at 65.2%. This higher prevalence likely reflects the sensitivity of the scoring system, enabling the measurement of 'lower-level' morbidity not captured in previous studies. However, the consistent association of such ePOMS-defined morbidity

**Table 4** Multivariable Cox regression model for postoperative time to discharge in a cohort of patients undergoing surgery for chronic subdural haematoma (n=531), pooled across m=40 imputed data sets

|  | HR | 95% CI | P value |
|---|---|---|---|
| Age (per year increase) | 0.99 | 0.99 to 1.00 | 0.114 |
| Sex (male vs female) | 1.09 | 0.90 to 1.33 | 0.351 |
| GCS 15 (vs any-other value) | 1.03 | 0.97 to 1.08 | 0.377 |
| Referred patient | 1.46 | 0.98 to 2.17 | 0.063 |
| ASA (per 1 level increase) | 0.90 | 0.77 to 1.06 | 0.217 |
| Admission complications | 1.06 | 0.99 to 1.12 | 0.093 |
| Airways disease (yes/no) | 1.03 | 0.79 to 1.34 | 0.819 |
| CVS disease (yes/no) | 0.96 | 0.77 to 1.18 | 0.676 |
| Anticoagulated on admission (yes/no) | 0.45 | 0.37 to 0.54 | <0.001* |
| Heart failure (yes/no) | 1.02 | 0.77 to 1.33 | 0.908 |
| mRS (per 1 level increase) | 0.99 | 0.91 to 1.09 | 0.885 |
| Baseline creatinine (per 20 µmol/L increase) | 0.98 | 0.94 to 1.01 | 0.242 |
| Cognitive concern (yes/no) | 0.71 | 0.58 to 0.87 | <0.001* |
| Operative time (per 10 min increase) | 0.99 | 0.91 to 1.01 | 0.357 |
| Postoperative complications (per 1 domain increase in ePOMS) | 0.61 | 0.55 to 0.68 | <0.001* |

HRs >1 indicate association with a more rapid time to discharge.
*Indicate statistical significance at the 5% level. Postoperative complications included as a time-dependent covariate (by day).
ASA, American Society of Anesthesiologists score; CVS, cardiovascular; ePOMS, Electronic Postoperative Morbidity Score; GCS, Glasgow Coma Scale; mRS, Modified Rankin Scale.

with LOS indicates that it is likely to be of clinical relevance. As EHR and standardised data collection become more widespread, the utilisation of phenotyping techniques, such as the ePOMS, will become increasingly important but, in the short term, create a degree of difficulty in allowing direct comparisons to the previously published literature.

### Utility of EHR in exploring perioperative care

The evidence from our study is that interrogation of the EHR data may be a valuable addition to the improvement process, allowing the extraction and analysis of complex events and the use of computable phenotypes. This approach enables the evaluation of previously unmeasured clinical interactions on patient outcomes at different stages of the patient journey. The importance of integrated care at a system level is widely recognised in the care of traumatic brain injury. Here, interventions and further research can be prioritised based on an understanding of the drivers of outcome in each phase of care.[39] The nature of the cSDH referral pathway and the inherent complexity of the patient cohort means simple interventions to improve outcome are unlikely to be successful and a similar 'system-wide' approach may be required. Studies, such as ours which examine hitherto unexplored events, will be essential for defining both the scope of the problem and areas for improvement.

In this study, we have evaluated the impact of potentially significant elements of intraoperative care for the first time in a cohort of patients with cSDH. Significant associations with LOS were not found. We defined covariates based on time outside optimal ranges, deviation from which is associated with adverse outcome in other surgical specialties,[15] or recommended as physiologically appropriate in more severe head injury.[30 40] Periods of time (10 min or more) spent at a MAP of <80 mm Hg are associated with end-organ complications in noncardiac surgery.[15] In the same setting, a 'dosing effect' is suggested, with shorter periods of time under lower MAP values associated with complications. Overall, 80 mm Hg was chosen for this project due to its recommendation in other settings of neurosurgical practice[40] but perhaps total time below this threshold, without considering a dosing effect may explain the lack of any observed association. Similarly, it may be that MAP targets should be individualised based on patient characteristics such as comorbidities, and exploring such interactions represent a potential area of future research.

### Limitations of EHR data

Utilising EHR-derived data has limitations, the majority of which are grounded in the availability of readily extractable data encapsulating key clinical characteristics such as radiological diagnoses or the specific events occurring during an operative procedure. The absence of information such as comorbidities required the use of novel phenotyping methods based on prescription medicines.[31] Although this methodology has been used in a large Australian database, the score has not previously, to our knowledge, been used in the surgical setting. Thus,

novel elements of our methodology including drug-inferred comorbidities and the ePOMS require robust external validation in a distinct data set despite apparent face validity. In larger cohorts, exploration of whether a weighted score (where different domains exert differing effects on LOS) could be explored. The need for such methods may vary by cohort and type of admission (eg, emergency vs elective). Hospital coding for SDH does not confidently delineate acute from chronic SDH,[41] and operative codes do not encapsulate the specific technique used. In order to maintain anonymity of individual cases, we were not able to explore this by extracting free-text radiographic or operative details. Such analysis cannot be readily applied in a scaleable manner to allow automated calculation from aggregate data. Limitations on case definition in this study were mitigated through linkage with the centre's neurosurgical referral database, where cases are contemporaneously coded by a neurosurgical clinician, including a distinction between acute and cSDH. In addition, cases identified through coding data were refined by excluding those with associated major traumatic injuries to minimise the inclusion of cases of acute SDH. Despite these measures, it is possible that some cases were incorrectly included.

## Conclusion

This study provides the first comprehensive evaluation of inpatient care for surgically treated cSDH using routinely collected data from an integrated EHR. Perioperative morbidity was common and associated with increased postoperative LOS. Preoperative anticoagulation, admission GCS, cognitive dysfunction and referral source were also associated with postoperative LOS. This study identifies key areas to explore in quality improvement initiatives, while providing a methodological framework relevant to neurosurgery, as well as other surgical and inpatient specialities.

**Author affiliations**
¹University Division of Anaesthesia, Department of Medicine, University of Cambridge, Cambridge, UK
²Healthcare Design Group, Department of Engineering, University of Cambridge, Cambridge, UK
³Department of Clinical Neurosurgery, University of Cambridge, Cambridge, UK
⁴NIHR Global Health Research Group on Neurotrauma, University of Cambridge, Cambridge, UK
⁵NIHR Brain Injury MedTech Co-operative, University of Cambridge, Cambridge, UK
⁶Academic Neurosurgery, University of Cambridge, Cambridge, UK
⁷Neurocritical Care Department and Department of Anaesthesia, Cambridge University Hospitals NHS Foundation Trust, Cambridge, UK

**Contributors** DJS: Concept, study design, data manipulation, primary analysis, manuscript preparation. BD: Study design, manuscript preparation and editing. TB: Study design, manuscript preparation and editing. AJ: Data access, study design, manuscript preparation and editing. PJH: Interpretation of findings, manuscript preparation and editing. DKM: Interpretation of findings, manuscript preparation and editing. AE: Statistical and data science advice and supervision, manuscript preparation editing. RMB: Data access, study design, supervision, manuscript preparation and editing.

**Funding** DJS is supported by a Wellcome Trust Clinician PhD Fellowship (overarching grant number: 204017/Z/16/Z).

**Competing interests** PJH is supported by the NIHR (Research Professorship, Cambridge BRC, Global Health Research Group on Neurotrauma) and the Royal College of Surgeons of England; DKM is supported by the NIHR (through an Emeritus Senior Investigator appointment and the Cambridge NIHR Biomedical Research Centre, and funding for Global Health Research Group on Neurotrauma). TB is supported by the NIHR Global Health Research Group on Neurotrauma at the University of Cambridge. Internal Reference: RG89187 AJJ is supported by the NIHR Brain Injury MedTech Co-operative based at the University of Cambridge. The views are those of the authors and not necessarily those of the NHS, the NIHR, the Wellcome Trust or the Department of Health.

**Patient and public involvement** Patients and/or the public were not involved in the design, or conduct, or reporting, or dissemination plans of this research.

**Patient consent for publication** Not required.

**Ethics approval** The study was performed as part of a local service improvement programme, with local service evaluation approval obtained in line with UK procedure (ref: PRN 7705).

**Provenance and peer review** Not commissioned; externally peer reviewed.

**Data availability statement** No data are available. The data were collected as part of routine care and accessed as part of an NHS service evaluation. As a result the authors are not in a position to make the data freely available. The authors will entertain scientific collaborations with interested parties on application subject to data use agreement by the trust.

**ORCID iDs**
Daniel J Stubbs http://orcid.org/0000-0003-2778-5226
Benjamin M Davies http://orcid.org/0000-0003-0591-5069
Tom Bashford http://orcid.org/0000-0003-0228-9779
David K Menon http://orcid.org/0000-0002-3228-9692
Ari Ercole http://orcid.org/0000-0001-8350-8093

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
