## [Reviewer comments · BMJ Open]

ARTICLE DETAILS

TITLE (PROVISIONAL)	Identification of factors associated with morbidity and post-operative length of stay in surgically managed chronic subdural haematoma using electronic health records: a retrospective cohort study
AUTHORS	Stubbs, Daniel; Davies, Benjamin; Bashford, Tom; Joannides, Alexis; Hutchinson, Peter; Menon, David; Ercole, Ari; Burnstein, Rowan

VERSION 1 – REVIEW

REVIEWER	Sam Hall University Hospitals Southampton, UK
REVIEW RETURNED	25-Feb-2020

GENERAL COMMENTS	Thank you for inviting me to review the manuscript titled "Utility of electronic health records in exploring associations between complex perioperative events morbidity and post operative length of stay in surgically managed chronic subdural haematoma" submitted to the BMJ by the team at Cambridge University Hospitals. The objective of this article is timely given the increasing use of electronic medical records. Electronic records make it practical to collect large volume data sets including granular data such as individual recordings of physiological parameters. Furthermore cSDH is a common neurosurgical condition whose incidence is likely to increase in our aging population thus is it advantageous to better understand the behaviour of these patients. This study is designed as a retrospective observational study using a single centre cohort of surgically managed patients with cSDH. The method of patient selection is appropriate, well described and illustrated with a clear flow-diagram. However the description of the final cohort would benefit from the addition of several additional details which may impact on outcomes and be of interest to a surgical audience. This includes the type of operation (burr-hole v mini-craniotomy), the use of subdural drains, mean operative time and grade of surgeon, which are presumably all available on an electronic health record. This article only includes patients operated under general anaesthesia. Do you perform all operations for cSDH under general anaesthetic or are there patients with severe co-morbidities and thus high anaesthetic risk who are operated under local anaesthetic and not included in this article?
--

	The authors describe the whole cohort in table 1 but the results and figure 3 describe length of stay for direct admissions v transferred patients – table 1 could be expanded with columns to describe these subgroups for comparison. The statistical methods are appropriate for the study. The measurement of complications was performed using the ePOMS score calculated daily. The POMS score was designed for major surgery and required external validation for neurosurgery (Grocott 2007). You should provide comment on its validity in neurosurgery as one would argue that the physiological stress from burr holes is different to that of major abdominal/orthopaedic surgery. The authors acknowledge the concern that ePOMS events may be triggered by clinically insignificant recordings. Do you have a protocol or local guideline on how often patients have their basic physiological parameters measured as more frequent recordings are more likely to identify clinically insignificant fluctuations in such parameters. Also, are all of the ePOMS domains, equal in their likelihood of increasing length of stay? For example is one episode of anti-emetic usage equal to a troponin rise? In the introduction cSDH is likened to neck of femur fracture as a sentinel health event. Is it possible for this exploratory analysis of electronic health records to comment on 6 or 12 month mortality rates in this cohort to support the statement? There are several typographical errors and first-person statements throughout the article which require proof-reading. Overall, this is an interesting article which demonstrates the feasibility of using electronic health records to measure peri-operative and post-operative factors, and use them to predict post-operative length of stay. I support this article for publication following minor revisions.
--	--

REVIEWER	Aladine Elsamadicy Yale School of Medicine, Department of Neurosurgery, USA
REVIEW RETURNED	02-Mar-2020

GENERAL COMMENTS	I commend the authors on exploring a common neurosurgical pathology that impacts many elderly patients, however I do have concerns with the clarity of the study that may be beneficial for the readers.  1) In the Abstract nor Methods is it clear about the surgical approach that was taken. Were these all burr holes for evacuation? Were there any acute on chronic subdurals that required a mini-craniotomy for evacuation given the septations? 2) How many patients were elective vs emergent? The timing of when the patient presents and goes to the OR an impact. If this is not known then needs to be in the limitations paragraph. 3) What would be more interesting would be the proportion of newly diagnosed complications each day, instead of a prevalence curve overtime, as these curves are difficult to extrapolate the meaning. 4) What was the grading scale used to determine cognitive impairment? If a patient has dementia at baseline, was this accounted for. 5) The interesting aspects of the study were the intraoperative variables, such that did the MAPs intraoperatively impact the
---

	proportion of post-operative MI? These were findings I was hoping to see or at least looked at. 6) The discussion needs to be cleaned up to be more clear with the points of interest. Would recommend having a clear theme for each paragraph, with clear statements and citations that either support or refute the authors claims. Would also recommend a paragraph suggesting what are ways providers can reduce complications associated with either intra-operative or post-operative adjustments. Overall I commend the authors on embarking a much needed subject, however there still needs to be work with the clarity of the writing as well as the data presented.
--	---

VERSION 1 – AUTHOR RESPONSE

Response to reviewer comments: Manuscript ID - bmjopen-2020-037385

Reviewer 1:

“...the description of the final cohort would benefit from the addition of several additional details which may impact on outcomes and be of interest to a surgical audience. This includes the type of operation (burr-hole v mini-craniotomy), the use of subdural drains, mean operative time and grade of surgeon, which are presumably all available on an electronic health record...”.

We thank the reviewer for this observation and agree that these factors are often considered in the context of outcomes from chronic subdural haematoma. Despite this data theoretically existing with an EHR, in our system it is stored in a non-structured fashion i.e. there is no specific box / form or code to discriminate these factors. For example, operation codes do not distinguish burr hole and mini craniotomy. As our study was designed to utilise aggregate, anonymised data to explore factors affecting post-operative outcomes, manual interpretation of free text data was not possible.

Whilst the suggested factors are often considered important, we note: (1) a meta-analysis has not identified outcome differences between mini-craniotomy and burr hole drainage (Almenawer et al 2014 Ann Surg 259(3):449-57) (2) the national UK survey of practice (which our centre participated in) indicated that 97% of all subdural surgery within the UK is performed by a non-consultant grade neurosurgeon (Brennan et al 2016) and (3) the insertion of subdural drains is part of the standard operating procedure at Cambridge. Whilst recognising this potential limitation we feel it is therefore unlikely that the absence of this information would alter interpretation of our key findings.

We have made changes within the manuscript to acknowledge this, including a section on standard surgical and postoperative practice within our centre (Lines 171 – 181) and included the median operation time at the start of the results section along with other values for our intraoperative covariates.

"This article only includes patients operated under general anaesthesia. Do you perform all operations for cSDH under general anaesthetic or are there patients with severe co-morbidities and thus high anaesthetic risk who are operated under local anaesthetic and not included in this article?"

We agree that this is an important consideration but were limited once again by the nature of the anonymous data which was extracted from the EHR. However, local anaesthesia for cSDH surgery is not routinely practiced at our institution except in very rare exceptional cases. We have now acknowledged the fact that it is possible that within the 'non-volatile anaesthesia reference group' some patients who had surgery under local anaesthesia could have been included (Lines 291-296).

Anaesthetic type was allocated into a broad 'volatile Y/N' term by the identification of non-zero values of inhalational anaesthetic recorded on a patients anaesthetic chart (235/530). As this data is automatically recorded it should be of high quality. A large majority of the remaining patients received an intravenous general anaesthetic (TIVA) (240/530). Unlike volatile anaesthetics the use of a TIVA anaesthetic must be manually entered by the attending anaesthetist, and inspection of the drug administration chart in 'non-volatile, non TIVA' patients revealed that many had received agents such as neuromuscular blockers indicating that they had received a general anaesthetic.

As cases under local anaesthetic may or may not have a monitoring anaesthetist and could conceivably receive medications such as opioids or sedatives to supplement local anaesthesia we have not attempted to infer the presence of local anaesthetic cases based on recorded medications. This point is included in our discussion as an example of the limitations of conducting work using extracted EHR data.

The authors describe the whole cohort in table 1 but the results and figure 3 describe length of stay for direct admissions v transferred patients – table 1 could be expanded with columns to describe these subgroups for comparison..

Thank you, we have modified Table 1 to the suggested format.

The measurement of complications was performed using the ePOMS score calculated daily. The POMS score was designed for major surgery and required external validation for neurosurgery (Grocott 2007). You should provide comment on its validity in neurosurgery as one would argue that the physiological stress from burr holes is different to that of major abdominal/orthopaedic surgery.

We agree with the reviewer that the physiological stress presented by major abdominal surgery is likely significantly different to that encountered during operative management of subdural haematoma. However, our results demonstrate an association between modified ePOMS morbidity and postoperative hospital stay, suggesting that there is apparent face validity. We recognise that this

does not substitute for external validation or a direct comparison to other relevant morbidity measures and we have amended our methods (lines 188 – 190) and discussion (lines 477) to reflect this.

The authors acknowledge the concern that ePOMS events may be triggered by clinically insignificant recordings. Do you have a protocol or local guideline on how often patients have their basic physiological parameters measured as more frequent recordings are more likely to identify clinically insignificant fluctuations in such parameters. Also, are all of the ePOMS domains, equal in their likelihood of increasing length of stay? For example is one episode of anti-emetic usage equal to a troponin rise?

We have now included a discussion of local policy on observation of physiological measurements, and the corresponding national guidance in the section on 'Local Surgical Practice' (lines 171-181). Escalation in observation frequency is intrinsically related to the identification of abnormalities within recorded values, therefore although we agree that more frequent observations increases the chance of a single observation being classed as 'abnormal' the pre-test probability of abnormality will also be increased (reflecting the underlying increase in observation frequency).

We agree with the reviewer that exploration of the contribution of individual ePOMS domains is important but have not pursued this in detail for several reasons. Firstly, for fair comparison our statistical model (with a time dependent measure of overall morbidity) would need to be run multiple times for each domain introducing concerns regarding multiple testing as well as low power in domains where few events occur on an ongoing basis – see also response to Reviewer 2, below re: incident complications). We agree this should be an area of ongoing work (line 478) but feel this would be best done in a larger dataset. Secondly, elements of the ePOMS score have already been included in other modelling (such as admission GCS). We have however included an estimate of the effect of reoperation (a rare component of our modified ePOMS score) but one that is very strongly associated with prolonged stay (HR 0.38 p <0.0001 – line 290).

In the introduction cSDH is likened to neck of femur fracture as a sentinel health event. Is it possible for this exploratory analysis of electronic health records to comment on 6 or 12 month mortality rates in this cohort to support the statement?

Whilst this is an interesting point, this was beyond the scope of our single centre evaluation study using electronic record data. In addition, access to records from external referring hospitals was not possible to allow us to answer this question and have highlighted this in the discussion (line 404). It is something we are actively exploring in further work.

There are several typographical errors and first-person statements throughout the article which require proof-reading.

We hope that these have now been addressed

Reviewer: 2

Reviewer Name: Aladine Elsamadicy

Institution and Country: Yale School of Medicine, Department of Neurosurgery, USA

Please state any competing interests or state 'None declared': None

I commend the authors on exploring a common neurosurgical pathology that impacts many elderly patients, however I do have concerns with the clarity of the study that may be beneficial for the readers.

1) In the Abstract nor Methods is it clear about the surgical approach that was taken. Were these all burr holes for evacuation? Were there any acute on chronic subdurals that required a mini-craniotomy for evacuation given the septations?

Unfortunately, as outlined in the response to reviewer 1, the non-structured ('free text') nature in which this information is recorded has prevented its extraction from our EHR. We recognise that this is an important limitation and, as outlined in our response to reviewer 1 above, have made changes to acknowledge this.

2) How many patients were elective vs emergent? The timing of when the patient presents and goes to the OR an impact. If this is not known then needs to be in the limitations paragraph.

We agree with the reviewer that the urgency of surgery is an important area of consideration. We do not have access to NCEPOD urgency codes (widely used in the NHS) as a marker of urgency. However, our local practice is for cSDH to be operated on an emergent basis. In addition, we feel that such a marker would likely reflect the physiological derangement induced by their pathology that is already captured in other covariates (e.g. abnormal motor score). We have however included a comparison of time to surgery from within our centre (Table 1) to provide context and highlight that this does not include the transfer time between centres for those referred from elsewhere.

Commented [Office1]: Do you not have a code for whether the admission was an emergency vs elective? Doubt there are any electives, and if so, you can cite the % in the rebuttal

3) What would be more interesting would be the proportion of newly diagnosed complications each day, instead of a prevalence curve overtime, as these curves are difficult to extrapolate the meaning.

We agree with the reviewer that this is an important consideration, as the 'flatness' of our prevalence curves could indicate either an equivalent rate of incident complications to replace people recovering (or leaving our population through discharge) or simply a recovery from baseline without ongoing new cases. We have included two figures within the supplementary material: one indicating neurological complications and another indicating non-neurological complications (excluding reoperation). This

suggests that neurological recovery is rapid with surgery preventing ongoing new complications but that there is ongoing development of new complications throughout inpatient stay. (Lines 312-318)

4) What was the grading scale used to determine cognitive impairment? If a patient has dementia at baseline, was this accounted for.

At the point of admission all patients are classified by admitting staff as having either temporary, permanent, or no cognitive impairment. Adoption of more formal measures of cognition or confusion (such as the 4AT or AMTS) have only been mandated recently at our centre and not available for the study cohort. Whilst recognising the limitations of such an approach, we therefore used a summary measure of 'any cognitive impairment temporary or permanent v no impairment' which should encompass prevalent cases of dementia and have highlighted this in our methods (lines 239-241)

5) The interesting aspects of the study were the intraoperative variables, such that did the MAPs intraoperatively impact the proportion of post-operative MI? These were findings I was hoping to see or at least looked at.

This is an active area of work for us and we are glad that the reviewer also feels it is of interest. However, our objectives with this manuscript were to present a comprehensive statistical and methodological approach to demonstrate the feasibility and utility of using routinely collected electronic record data to study peri-operative morbidity. Thus, in order to preserve the clarity of message in this study, we feel an in-depth analysis of individual variables is beyond the scope of this manuscript.

6) The discussion needs to be cleaned up to be more clear with the points of interest. Would recommend having a clear theme for each paragraph, with clear statements and citations that either support or refute the authors claims. Would also recommend a paragraph suggesting what are ways providers can reduce complications associated with either intra-operative or post-operative adjustments.

We have now included sub-headings within the discussion to clarify the important points of interest, and made new inclusions as detailed above to address the reviewers' concerns.

VERSION 2 – REVIEW

REVIEWER	Aladine Elsamadicy Yale School of Medicine
REVIEW RETURNED	21-Apr-2020
GENERAL COMMENTS	Authors have answered my inquires in a satisfactory manner.